# An evaluation of interventions within a Growth Through Nutrition project aimed at enhancing optimal nutrition and water, sanitation and hygiene (WASH) and nutrition practices among nutritionally most vulnerable households (MVHHs) in Ethiopia

Cherinet Abuye[1]*, Daniel Abbott[1], Lioul Berhanu[1], Adam Bailes[2], Rachel Holtzman[2]

1 Growth Through Nutrition, Save the Children International, Addis Ababa, Ethiopia, 2 Save the Children US, Fairfield, CT, United States of America

* cherinetabuye1@yahoo.com

## Abstract

Despite a downward trend, Ethiopia still faces significant challenges with high rates of stunting and acute malnutrition in children. To tackle these issues, the Feed the Future Ethiopia Growth Through Nutrition Activity, a USAID-funded project aligned with Ethiopia's National Nutrition Program, was executed from 2016 to 2023. This initiative aimed to enhance nutrition for women and young children across six regions through multisectoral interventions. Annual surveys conducted in 2017, 2018, and 2020 evaluated the impact of livelihood support and Water, Sanitation, and Hygiene (WASH)-focused social behavior change communication (SBCC) on vulnerable households with children under two. The results showed significant improvements in child nutrition. Dietary diversity among children increased from 12% to 34% (p<0.001), and the percentage of children receiving a minimal acceptable diet rose from 12% to 30.7% (p<0.001). Women's dietary diversity also improved markedly, from 2% to 16% (p<0.001). Handwashing practices saw improvements, with the proportion of households having handwashing facilities rising from 14% to 31% (p<0.001), and the adherence to critical handwashing moments increased from 16% to 23%. However, challenges in water treatment and latrine use persisted, with less than 25% improvement. The findings suggest that integrating livelihood support with SBCC interventions can positively enhance nutritional outcomes. Continued focus on these strategies is recommended to further support vulnerable households.

## Introduction

The world is falling short of achieving the Sustainable Development Goals (SDGs) related to reducing malnutrition. As of 2022, an estimated 22.3% of children under five globally were

**Data Availability Statement:** All relevant data are within the manuscript and its Supporting Information files. The data is provided in an Excel file named "S1 Data".

**Funding:** The author(s) received no specific funding for this work.

**Competing interests:** The authors have declared that no competing interests exist.

stunted and 6.8% were wasted. Low- and middle-income countries, particularly in Sub-Saharan Africa and South Asia, bear the heaviest burden. Often coexisting chronic food shortages, inadequate childcare, illnesses, and poor water, sanitation, and hygiene (WASH) all contribute to chronic child undernutrition in these regions [1, 2].

Given the complex causes of malnutrition, a coordinated response across various sectors is conceptually accepted as necessary. Recent studies show the effectiveness of multisectoral interventions in improving nutritional practices and the nutritional status of children and mothers. Programs that address food insecurity, WASH issues, and childcare practices comprehensively have demonstrated positive impacts on infant and young child feeding, hygiene and sanitation practices, and even nutritional status of children in low-income countries [2–8]. Similarly, in rural and agrarian communities, agricultural diversity encourages the consumption of locally produced foods and increases income for low-income households, ultimately leading to dietary diversity [9–12]. Combined with access to safe water and proper hygiene and sanitation practices, these actions help reduce waterborne illnesses and improve nutrition and health outcomes, especially among vulnerable groups like infants, young children, and pregnant or lactating women [4, 8, 13].

As of 2019, 37% of under-five children in Ethiopia were affected by stunting, and 7% suffered from wasting. Major contributors to these high rates include food insecurity, poor diet quality, insufficient food intake, lack of nutritional knowledge, and poor hygiene and sanitation. Stunting was more prevalent in rural households (42%) and those in the lowest economic bracket (42%) compared to the highest (24%) [14]. Recognizing the complex causes of malnutrition, the Ethiopian National Nutrition Program (2016–2020) emphasized a multisectoral approach to curb its burden [15].

In alignment with local needs and the Ethiopian National Nutrition Program framework, USAID launched the Feed the Future Ethiopia program, Growth through Nutrition Activity, in 2016. This initiative aims to enhance the nutritional status of women and young children in rural districts across six agrarian regions. The program employs a multisectoral approach, integrating efforts across health, agriculture, water, and sanitation, with a strong emphasis on social behavior change communication (SBCC). Funded by USAID and executed at district and local levels, the project aims to benefit over 14 million people across the six regions.

The Growth through Nutrition project focused on supporting the most economically vulnerable households "poorest of the poor households", identified by local committees. These households had a child under two years old and faced challenges in meeting basic needs due to resource limitations or unfavorable living conditions. The project provided livelihood support, including productive animals like sheep, goats, and chickens, seeds of nutritious vegetables and fruit seedlings, and basic gardening tools. This support was complemented by training on the production, consumption, and promotion of these nutrient-dense foods. Additionally, the project promoted optimal maternal, infant, and young child feeding practices, encouraged the use of nutrition services such as iron-folate supplementation for pregnant women, and advocated for improved hygiene and sanitation through SBCC interventions. Efforts to increase access to safe water were targeted at the community level, while household interventions focused on promoting hygienic practices such as handwashing and proper latrine construction and use. Although water supply schemes were included in the intervention, they were not exclusive to the population under study. Consequently, we did not analyze the households' water access.

The project team conducted three assessments on cohorts of MVHHs to measure the impact of targeted interventions on dietary and WASH-related household practices. The primary objective of this study is to share the aggregated findings of these assessments, which evaluated changes in dietary and WASH-related household practices associated with the livelihood and WASH-related interventions undertaken by the project.

## Study approach and methodology

### Study area and approach

This section describes the methodology of a cohort panel study conducted among rural MVHHs participating in the Growth through Nutrition Project Intervention in Ethiopia. The study included all intervention regions and assigned unique identification numbers to participating households. Baseline and follow-up monitoring assessments were conducted on Oct 10-Nov 30, 2017 (baseline), Oct 20 –Nov 4, 2018 (midline), and Feb 24 –March 6, 2020 (final). Enumerators aimed to interview the same respondent in each MVHH during each year of data collection. Surveys were repeated annually with the same cohort of households and additional households from neighboring kebeles added in year three. In year three, as households with infants or young children under 2 years of age changed within the original cohort due to aging out of the study units, comparable households from neighboring kebeles were added in the sample. The same questions, measurement tools, and data collection procedures were used throughout the study.

### Sampling method and sample size

This household study was conducted among 4,000 year-one project supported MVHHs in 250 kebeles across five regions of Ethiopia. A total of 400 MVHHs were selected proportionally to their representation in each region's kebeles (lowest administrative units in Ethiopia). The study involved MVHHs from 25 kebeles and covered all 16 MVHHs in each selected kebele. The sample size was calculated using a conventional sample size estimation procedure, resulting in a recommended sample size of 351 households. However, due to the study's panel nature, which involves interviewing the same households throughout the study, a 10% attrition rate was estimated, resulting in a total of 400 MVHHs being interviewed.

### Ethical considerations

The study obtained ethical approval from SCUS' Ethics Review Committee. The data collection team informed district/ woreda authorities and village leaders about the study's objectives and obtained oral consent. The team also obtained informed consent from all respondents and ensured their confidentiality and privacy. The respondents provided their consent verbally, and data were collected and stored anonymously while the questionnaire was administered confidentially.

### Data collection and data quality assurance

The study employed pre-tested questionnaires to gather household data and employed trained and experienced data collectors and supervisors who possessed sufficient and relevant field experience, as well as knowledge of local languages. In each region, two data collectors and one supervisor obtained permission from district administrators and conducted interviews door-to-door with community guides. Data was collected via tablets using KoBo Toolbox, applying local language translated and pre-tested tools, and was closely monitored by supervisors and MEAL managers to ensure adherence to the study protocol.

### Data analysis

The Monitoring and Learning/Evaluation team of Growth through Nutrition in Ethiopia supervised data entry and cleaning before handing over to the Senior Research and M&E Advisor at Save the Children US (SCUS) for data analysis using SPSS software. Statistical tests such as chi-square, McNemar's test, and t-tests were used to assess significance in changes for

select nutritional outcome indicators of interest between the baseline and follow-up assessments. Different statistical tests were utilized during data analysis depending on the indicator being examined across the cohorts. For most indicators, statistical analysis was conducted comparing only Cohort 1 data in 2020 to baseline data in order to assess changes within the same households. The findings are reported in descriptive percentages, means, sum-totals and p-values. A p-value less than 0.05 was considered statistically significant.

## Results

### Household characteristics

Table 1 indicates that over 94% of the respondents in the cohorts were women aged between 18 and 55 years, with a mean age of 32 years. The majority of the respondents (80%) were married to the head of the household, and 61% had no education. The average household size was 5.5 members in Cohort 1 and 5.2 members in Cohort 2. In the new cohort, Cohort 2, the proportion of households with married couples was 80%.

### Support provided by project to MVHHs by type

All households (100%) received livestock support, while 91% received farm tools, and 95% received seeds (Table 2). The majority of households, over 90%, found this support helpful and relevant to their needs. To further support agricultural activities, the project provided training and technical assistance to MVHH residents. The percentage of households participating in training significantly increased from 30% in 2018 to 94% in 2020. Additionally, the percentage of households receiving technical support rose from 13% in 2018 to 67% in 2020, indicating that the project's technical support reached MVHHs effectively.

In addition to agricultural support, many MVHHs participated in other services offered by the project. For instance, the percentage of households organized into village saving and credit groups increased from 78% in 2018 to 94% in 2020, while participation in monthly sessions of community conversations on nutrition and WASH, labeled by the project as Enhanced

**Table 1. Background characteristics of household respondents among sampled MVHHs, (2017–2020).**

| Respondent Characteristics | | Baseline 2017 | Follow-Up 2018 | Follow Up 2020 All Cohort | Follow Up 2020 Cohort 1 | Follow Up 2020 Cohort 2 |
|---|---|---|---|---|---|---|
| | | % (n) | % (n) | % (n) | % (n) | % (n) |
| Gender | Female | 100 (386/386) | 99.2 (354/357) | 94.4 (619/656) | 94.4 (301/319) | 94.4 (318/337) |
| | Male | 0.0 (0/386) | 0.8 (3/357) | 6.6 (37/656) | 5.6 (18/319) | 5.6 (19/337) |
| Age in years | Mean | 30.8 (386) | 31.6 (357) | 32.2 (621) | 32.7 (303) | 31.7 (318) |
| Relation to head of household | Head of household | 15.5 (60/386) | 15.2 (54/357) | 14.5 (95/656) | 14.7 (47/319) | 14.2 (48/337) |
| | Wife/ husband | 81.1 (313/386) | 80.6 (287/357) | 78 (512/656) | 78.4 250/337) | 77.7 (262/337) |
| HH size | Mean | 5.4 (386) | 5.5 (357) | 5.3 (656) | 5.5 (319) | 5.2 (337) |
| Marital status | Never married | 0.5 (2/386) | 0.5 (2/357) | 0.8 (5/656) | 1.3 (4/319) | 0.3 (1/337) |
| | Married/ living together | 85.0 (328/386) | 84.0 (300/357) | 80 (525/656) | 80.6 (257/319) | 79.5 (268/337) |
| | Divorced/ Separated/ Widow | 14.5 (56/386) | 15.4 (55/357) | 19.2 (126/656) | 18.1 (58/319) | 20.2 (68/337) |
| Level of education | No education | 65.8 (254/386) | 66.4 (237/357) | 63.7 (418/656) | 65.8 (210/319) | 61.8 (208/337) |
| | Adult education/literate | 1.6 (6/386) | 0.6 (2/357) | 0.9 (6/656) | 0.9 (3/319) | 0.9 (3/337) |
| | Preschool | 0.3 (1/386) | 0.3 (1/357) | 0.3 (2/656) | 0.3 (1/319) | 0.3 (1/337) |
| | Primary | 27.5 (106/386) | 26.9 (96/357) | 31.3 (205/656) | 28.8 (92/319) | 33.5 (113/337) |
| | Secondary | 4.4 (17/386) | 5.9 (21/357) | 3.7 (24/656) | 3.8 (12/319) | 3.6 (12/337) |
| | Technical/ vocational | 0.5 (2/386) | 0.0 (0/357) | 0.2 (1/656) | 0.3 (1/319) | 0 (0/337) |

**Table 2. MVHHs received support from the project by type of support, (2017–2020).**

| Services or support provided by the project (yes) | 2017 | 2018 | 2020 All Cohort | 2020 Cohort 1 | 2020 Cohort 2 |
|---|---|---|---|---|---|
| | % (n) | % (n) | % (n) | % (n) | % (n) |
| Livestock support* | 94.8 (366/386) | 91.9 (328/357) | 99.7 (654/656) | 99.4 (317/319) | 100 (337/337) |
| Was support helpful and/or relevant? | 99.5 (364/366) | 87.5 (287/328) | 92.4 (604/654) | 90.5 (287/317) | 97 (317/337) |
| Farm tools support** | 73.8 (285/386) | 19.0 (68/357) | 90.5 (594/656) | 91.2 (291/319) | 89.9 (303/337) |
| Was helpful/relevant | 97.5 (278/285) | 94.1 (66/68) | 96.3 (572/594) | 95.5 (278/291) | 97 (294/303) |
| Seed/Seedling support*** | 83.9 (324/386) | 86.3 (308/357) | 95.1 (624/656) | 94.4 (301/319) | 95.8 (323/337) |
| Was support helpful and/or relevant? | 94.1 (305/324) | 88.6 (273/308) | 91.7 (572/624) | 90.7 (273/301) | 92.6 (299/323) |
| Training | 73.8 (285/386) | 30.0 (107/357) | 93.9 (616/656) | 92.8 (296/319) | 95 (320/337) |
| Was training helpful/ relevant | 100.0 (285/285) | 96.3 (103/107) | 95.5 (588/616) | 93.2 (276/296) | 97.5 (312/320) |
| Technical support (TA) on agricultural activities | 30.6 (118/386) | 13.4 (48/357) | 67.2 (441/656) | 66.8 (213/319) | 67.7 (228/337) |
| Was TA helpful/relevant | 100.0 (118/118) | 100.0 (48/48) | 90.7 (400/441) | 87.3 (186/213) | 93.9 (214/228) |
| Organized in Saving groups | 63.7 (246/386) | 78.2 (279/357) | 93.9 (616/656) | 94 (300/319) | 93.8 (316/337) |
| Was helpful/relevant | 99.6 (245/246) | 92.5 (258/279) | 89.8 (553/616) | 88.7 (266/300) | 90.8 (287/316) |
| Participated in ECC Sessions | 0.0 (0/386) | 54.9 (196/357) | 86.9 (570/656) | 89.3 (285/319) | 84.6 (285/337) |
| Was helpful/relevant | — | 96.9 (190/196) | 93.9 (535/570) | 92.3 (263/285) | 95.4 (272/285) |
| Nutrition counseling | 67.4 (260/386) | 29.2 (104/357) | 79.1 (519/656) | 77.7 (248/319) | 80.4 (271/337) |
| Was helpful/relevant | 99.6 (259/260) | 95.3 (101/104) | 98.1 (509/519) | 98.8 (245/248) | 97.4 (264/271) |
| Access to basic drinking water | 0.0 (0/386) | 2.8 (10/357) | 30.5 (200/656) | 29.5 (94/319) | 31.5 (106/337) |
| Was helpful/relevant | — | 100.0 (10/10) | 93.5 (187/200) | 93.6 (88/94) | 93.4 (99/106) |
| Access to basic sanitation services | 0.0 (0/386) | 19.9 (71/357) | 48.2 (316/656) | 49.2 (157/319) | 47.2 (159/337) |
| Was helpful/relevant | — | 100.0 (71/71) | 95.9 (303/316) | 94.3 (148/157) | 97.5 (155/159) |

*Two female sheep for sheep rearing; or two female goats for goat rearing and six chickens total (five pullets and one cockerel of two-month-old chicken)

**Three basic hand tools for homestead garden cultivation

***five fruit seedlings

Community Conversation (ECC) Sessions, rose from 55% in 2018 to 87% in 2020. Access to basic sanitation services also improved, with the percentage of households receiving support increasing from 20% in 2018 to 48% in 2020.

### Program exposure and feeding during pregnancy among households with a currently pregnant woman or a pregnant woman in the last six months

Among the households interviewed in 2020, 28% (181 out of 656) had a woman who was either currently pregnant or had been pregnant within 6 months prior to data collection (Table 3). Of those households with a currently or recently pregnant woman, 66% reported having received information about care during pregnancy in 2020, which marks an increase from the 47% of households that reported receiving such information in 2018. In 2020, the primary sources of information about pregnancy care were health extension workers (87% of respondents), health workers (46%), and enhanced community conversations (ECC) and MVHH or Saving Group Meetings (24%). Compared to 2018, a greater diversity of sources were reported in 2020, with only 22% of households having reported receiving information from a health worker and 13% from MVHH or Savings Group Meetings.

It is noteworthy that while the percentage of households receiving information from health workers increased in 2020, the percentage of households that received information through agricultural extension workers (AEWs) decreased from 50% in 2018 to only 22% in 2020 whereas, the contribution of the volunteer community network known as the Health

**Table 3. Sources of information for households with current or recently pregnant women, (2017–2020).**

| Did you or anyone in household receive information in the following topics in the past six months? | | 2017 | 2018 | 2020 |
|---|---|---|---|---|
| | | % (N) | % (N) | % (N) |
| Households with a woman currently pregnant or was pregnant in the last six months | | 59.6 (230/386) | 27.5 (98/357) | 27.6 (181/656) |
| Received information on taking care of pregnant mother during pregnancy | | 74.3 (171/230) | 46.9 (46/98) | 65.7 (119/181) |
| Source | Health Worker | 43.9 (75/171) | 21.7 (10/46) | 45.5 (54/119) |
| | Health Extension Worker | 76.0 (130/171) | 89.1(41/46) | 86.6 (103/119) |
| | Agriculture Extension Workers (AEWs) | 37.4 (64/171) | 50.0 (23/46) | 21.8 (26/119) |
| | HDA/WDA(Health Development Army/Women Development Army)* | 9.4 (16/171) | 0.0 (0/46) | 8.4 (10/119) |
| | Enhanced Community Conversation | 0.0 (0/171) | 41.3 (19/46) | 24.4 (29/119) |
| | MVHHs or Saving Group Meetings | 2.3 (4/171) | 13.0 (6/46) | 23.5 (28/119) |
| | Cooking Demonstration | 21.1 (36/171) | 0.0 (0/46) | 8.4 (10/119) |
| | Others (Religious Group Meetings, Radio Mobile Phone, Poster/Flyers/Leaflets etc.) | 0.5 (1/171) | 1.9 (1/46) | 1.7 (2/119) |

* Community volunteers providing information on health, nutrition, gender, etc. to a group of households in their neighborhoods

Development Army (HAD) as an information source remained low at around 9%. In terms of practices that respondents changed as a result of this information, 92% of respondents reported adopting at least one nutritional practice in 2020, a significant increase from 67% of respondents in 2018. In 2020, most respondents reported diversifying their food intake during pregnancy (79%), increasing the number of meals during pregnancy (70%), and taking iron folate tablets during pregnancy (61%). Compared to 2018 and the baseline (2017), there was an increase in the number of households that reported improving these three nutritional practices.

Overall, WASH practices such as treating household water and using improved latrines remained a challenge among households in 2020, with less than 25% of households reporting improvements in these areas. However, hand washing with water and soap improved from 28% in 2018 to 51% in 2020 (Fig 1).

## Program exposure and changes in breastfeeding practices among households with children under 2 years of age

Among the households surveyed in 2020, 87.3% reported receiving information about breastfeeding, increasing from 58% in 2018. The majority of women received this information from health extension workers (85%) or health workers (47%) in 2020 (Table 4). As for project interventions, in 2020 ECCs and MVHH or Saving Group Meetings provided information to 39% and 26% of households, respectively, which was consistent with the 2018 findings.

Mothers exposed to these sources of information reported improving at least one feeding practice in 96% of cases in 2020. Notably, in 2020 56% of households initiated breastfeeding within one hour, a notable increase from 42% in 2017. Additionally, 80% of households practiced exclusive breastfeeding for six months and continued breastfeeding in 2020 (Fig 2).

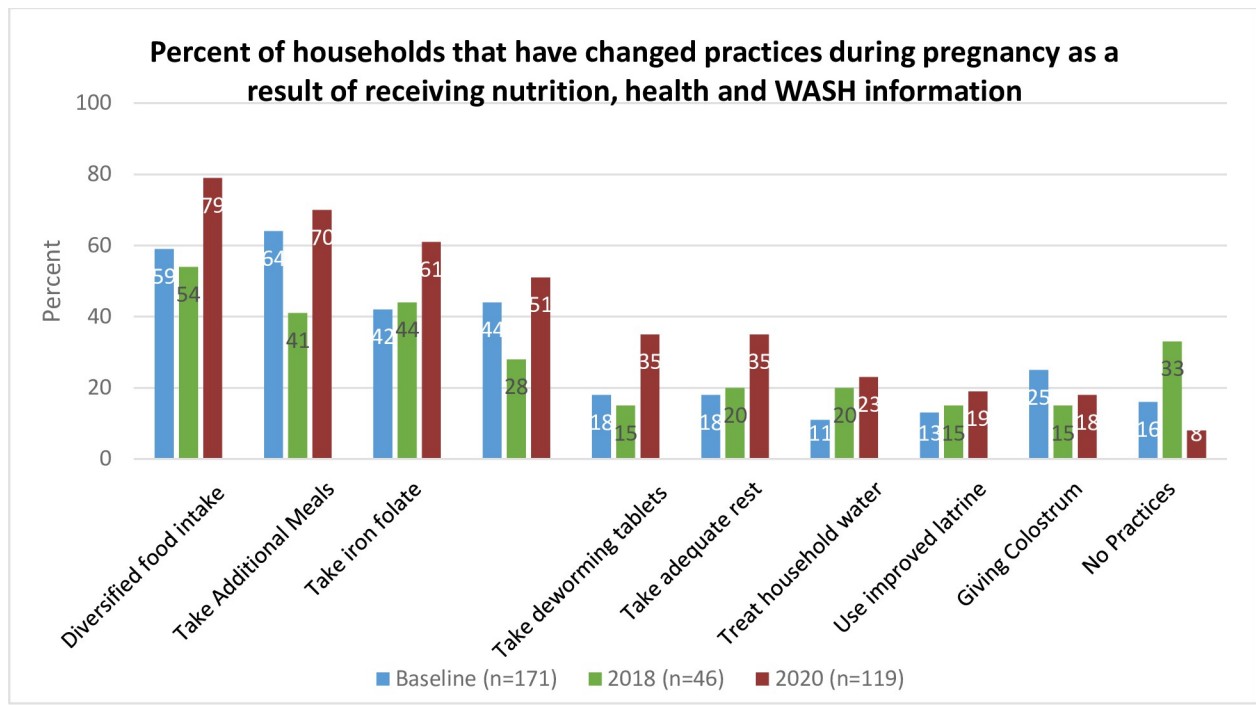

**Fig 1. Percent of households with recent or currently pregnant women that reported changing practices during pregnancy as a result of receiving nutrition, health and WASH information (2017–2020).**

Colostrum was given in 30% of households, and 29% avoided bottle feeding in 2020, up from 21% and 12% in the baseline, respectively.

## Program exposure and complementary feeding practices among households with children 6–23 months of age

Table 5 shows that out of the 104 households with children aged 6–23 months, 85.6% received information on dietary diversity in 2020, which was a slight decrease from the baseline (89.6%), but an increase from 2018 (66%). In 2020, most households received this information through health extension workers (81%) or health workers (45%). Enhanced community conversations, one of the Growth through Nutrition interventions, were also a common source of

**Table 4. Sources of information on breastfeeding for households with children under 2 years of age, (2017–2020).**

| Did you or anyone in household receive information in the following topics in the past six months? | | 2017 | 2018 | 2020 |
|---|---|---|---|---|
| | | % (N) | % (N) | % (N) |
| Households with children under 2 years old residing in the household. | | 69.4 (268/386) | 48.5 (173/357) | 23.9% (157/658) |
| Received information on breastfeeding | | 54.9 (189/268) | 58.4 (101/173) | 87.3 (137/157) |
| Source | Health Worker | 43.9 (83/189) | 5.9 (6/101) | 46.7 (64/137) |
| | Health Extension Worker | 75.7 (143/189) | 90.1 (91/101) | 84.7 (116/137) |
| | AEWs | 49.17 (94/189) | 35.6 (36/101) | 18.2 (25/137) |
| | HDA/WDA | 11.6 (22/189) | 1.0 (1/101) | 5.8 (8/137) |
| | Enhanced Community Conversation | 0.0 (0/189) | 41.6 (42/101) | 38.7 (53/137) |
| | Cooking Demonstration | 10.1 (19/189) | 1.0 (1/101) | 9.5 (13/137) |
| | MVHHs or Saving Group Meetings | 4.8 (9/189) | 26.7 (27/101) | 26.3 (36/137) |

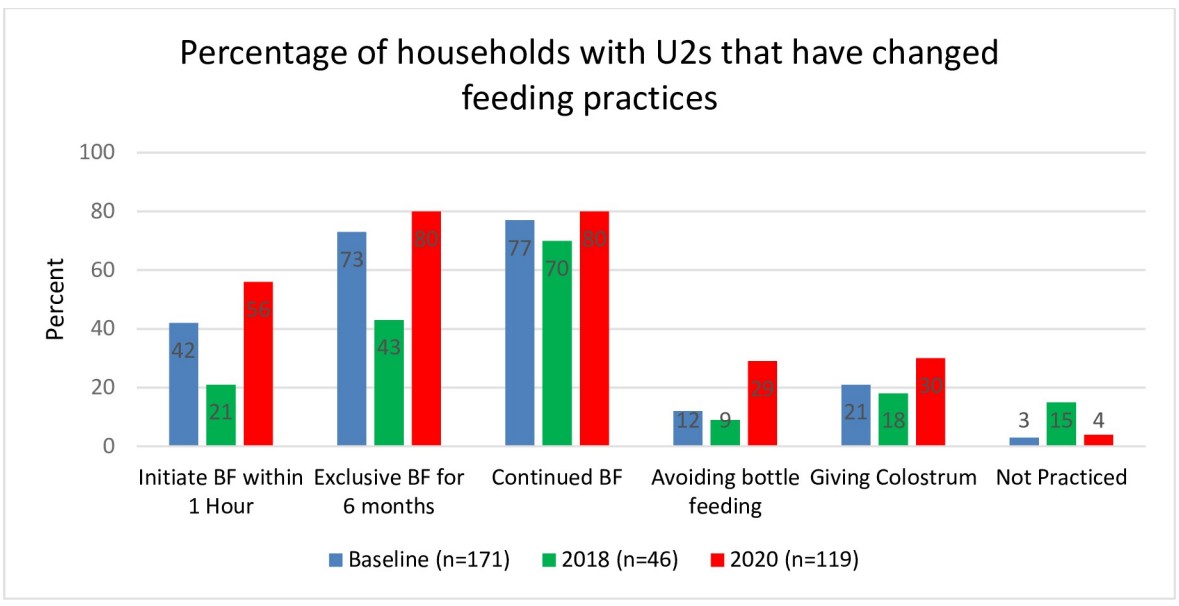

**Fig 2. Percentage of households with children under two that have changed feeding practices (2017–2020).**

information, but the proportion of households that identified it as a source decreased from 53.6% in 2018 to 42% in 2020. Meanwhile, MVHH or Savings Group Meetings have become a more popular source of information for households, rising from 8% and 18% in 2017 and 2018 to 28% in 2020 (Table 5).

In terms of feeding practice changes, most mothers reported adopting three key practices. Of households with children aged 6–23 months, 83% reported feeding their children diversified diets in 2020, up from 73% in the baseline. Sixty-four percent of households reported increasing feeding frequency in 2020, up from 58% in the baseline. Nominal improvements were seen in thick porridge consumption, with an increase from 51% in the baseline to 56% in 2020.

**Table 5. Sources of information for and reported changes in complementary feeding practices among households with children 6–23 months of age, (2017–2020).**

| Did you or anyone in household receive information in the following topics in the past six months? | | 2017 | 2018 | 2020 |
|---|---|---|---|---|
| | | % (N) | % (N) | % (N) |
| Households with children 6–23 months of age residing in the household. | | 66.1 (201/386) | 41.5 (148/357) | 15.8 (104/658) |
| Received information on dietary diversity for children 6–23 months | | 89.6 (180/201) | 65.5 (97/148) | 85.6 (89/104) |
| Source | Health Worker | 34.4 (62/180) | 7.2 (7/97) | 44.9 (40/89) |
| | Health Extension Worker | 68.3 (123/180) | 84.5 (82/97) | 80.9 (72/89) |
| | Agriculture Extension Workers (AEWs) | 61.1 (110/180) | 34.0 (33/97) | 19.1 (17/89) |
| | HDA/WDA | 12.8 (23/180) | 2.1 (2/97) | 9.0 (8/89) |
| | Enhanced Community Conversation | 0.0 (0/180) | 53.6 (52/97) | 41.6 (37/89) |
| | Cooking Demonstration | 6.7 (12/180) | 5.2 (5/97) | 9.0 (8/89) |
| | MVHHs or Saving Group Meetings | 7.8 (14/180) | 17.5 (17/97) | 28.1 (25/89) |
| Practices changed | Feeding your child diversified foods (4+ groups) | 72.8 (166/180) | 66.0 (64/97) | 83.1 (74/89) |
| | Give children >6 months animal sourced foods | 72.2 (130/180) | 46.1 (53/97) | 59.6 (53/89) |
| | Increase frequency of feeding | 58.3 (105/180) | 38.1 (37/97) | 64.0 (57/89) |
| | Thick Porridge | 50.6 (91/180) | 56.7 (55/97) | 56.2 (50/89) |
| | Not Practiced | 8.9 (16/180) | 11.3 (11/97) | 3.4 (3/89) |

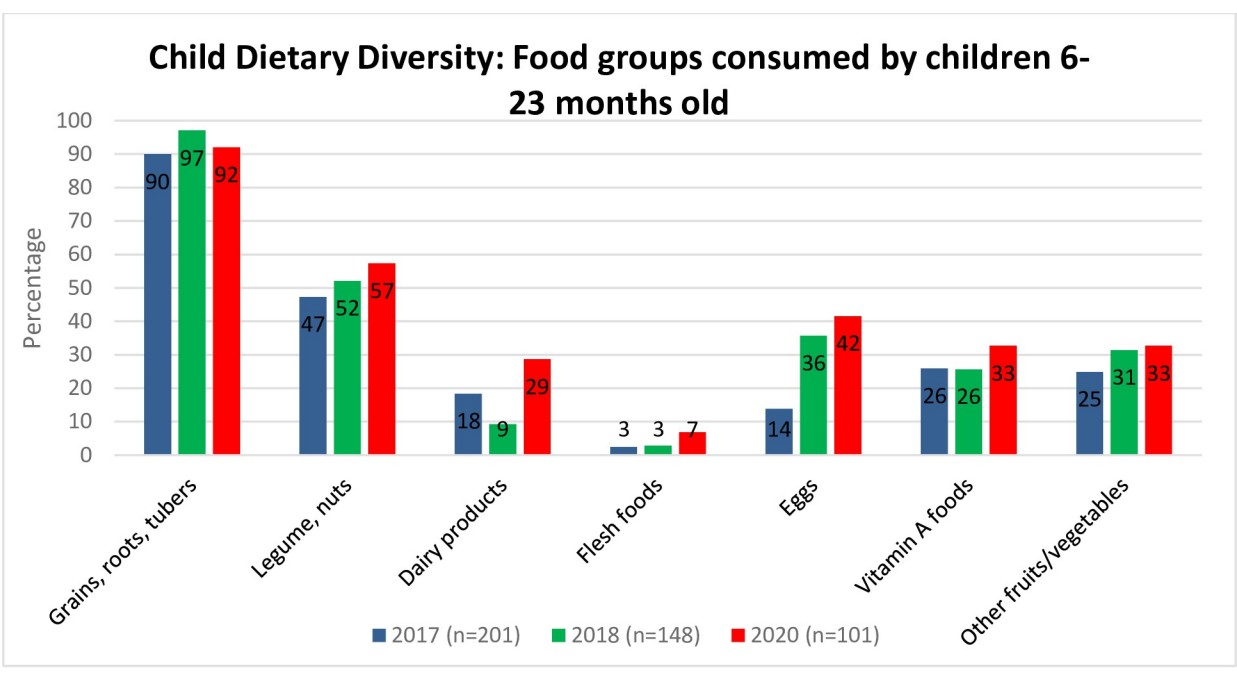

**Fig 3. Consumption of seven food groups by children 6–23 months old in MVHHs (2017–2020).**

### Food groups consumed by children 6–23 months old

Compared to the baseline, there was an increase in the percentage of children 6–23 months of age consuming all major food groups (Fig 3). The consumption of eggs showed the greatest improvement, with only 14% of children consuming eggs in the baseline, whereas in 2020, 42% of children consumed eggs. Moreover, there were increases in the percentage of children who consumed dairy products (18% in 2017 to 29% in 2020), legumes and nuts (47% to 57%), other fruits/vegetables (25% to 33%), and Vitamin A foods (26% to 33%). The percentage of children who consumed grains, roots, and tubers also slightly increased from 90% in the baseline to 92% in 2020. The project interventions were directly responsible for the increase in these food groups as they provided poultry, legumes, tubers, and nutrient-dense vegetable seeds along with support for their cultivation, which were the main components of the project's MVHH strategy. Furthermore, the consumption of dairy products increased due to the increase in dairy-producing livestock, such as heifers, goats, and sheep, that the MVHHs are raising. While the consumption of flesh foods slightly increased, it remains relatively rare among MVHHs as their livestock was mainly used for income generation rather than for feeding families. This is also evident from the fact that meat was still not commonly fed to children, and the affordability of meat remains a significant factor in low rates of meat consumption among children implying the need for continued economical and behavior- change related support.

### Infant and Young Child Feeding (IYCF)

The World Health Organization (WHO) recommends specific infant and young child feeding (IYCF) practices, including exclusive breastfeeding in the first 6 months of life, continued breastfeeding through age 2 years, and the introduction of solid and semisolid foods at age 6 months [16, 17]. The WHO also suggests that young children consume a diverse diet from different food groups, including animal source foods, to satisfy their growing micronutrient

needs. In this study, primary indicators of interest were indicators on breastfeeding (early initiation of breastfeeding, exclusive breastfeeding), dietary diversity, and meal frequency. Due to the smaller sample size of households with children under 2 in 2018, data from 2020 will be compared to baseline results to understand trends in behaviors and child outcomes.

Early initiation of breastfeeding is crucial for child health and nutrition as colostrum is highly nutritious and has antibodies that protect the newborn from diseases [16]. It is recommended that children be put to the breast within 1 hour after birth, and pre-lacteal feeding is highly discouraged. In our 2020 survey, only 66% of households with children under 6 months (36 out of 55) reported early initiation of breastfeeding (Table 6), decreasing from 73.1% in 2017. The decreasing prevalence of early breastfeeding initiation indicates that household and community-level activities to promote breastfeeding among MVHHs need to be strengthened to better support breastfeeding mothers.

Breast milk contains all the nutrients needed by children in the first 6 months of life, is an uncontaminated nutritional source, and reduces the risk of morbidity and mortality due to diarrhea and pneumonia [18–20]. It is recommended that children should be exclusively breastfed, consuming no other foods, during the first 6 months of their life. Among households with infants less than 6 months old, 93% reported exclusively breastfeeding their children in 2020. This has slightly decreased from the baseline, when 98.5% of respondents reported exclusively breastfeeding their children.

Children achieve minimum dietary diversity if they have consumed foods from at least four out of the seven food groups [21]. Achieving minimum dietary diversity of an infant or young child indicates micronutrient sufficiency (World Health Organization 2010b). Table 6 shows the percentage of children who consumed four out of the seven food groups increased from 12% in 2017 to 34% in 2020, a statistically significant increase (p<0.001). These results correlate with the observed increase in consumption across all seven food groups described in the previous section.

**Table 6. Changes in optimal infant and young child feeding practices among children under two years of age in MVHHs, (2017–2020).**

| Child-level outcomes | | 2017 | 2018 | 2020 | P-value^ |
|---|---|---|---|---|---|
| **Early initiation of breastfeeding** under six months | | 73.1% (49/67) | 62.5% (5/8) | 66.5% (36/55) | 0.439 |
| **Exclusive breastfeeding** under six months | | 98.5% (66/67) | 87.5% (7/8) | 92.7% (51/55) | 0.109 |
| **Mean # of food groups** from a maximum of 7 food groups for children 6 to 23 months of age | | 2.2 (201) | 2.2 (140) | 2.9 (101) | |
| **Child minimum dietary diversity** (4 food groups or more for children 6 to 23 months) | | 12.4% (25/201) | 18.9% (28/140) | 34.0% (34/100) | 0.000* |
| **Minimum meal frequency** (Child receives solid, semi-solid, or soft foods (but also includes milk for non-breastfed children) the minimum number of times or more over the previous day) | 6 to 8 months breastfed | 70.3% (26/37) | 96.2% (25/26) | 66.7% (14/21) | 0.776 |
| | 9 to 23 months breastfed | 73.6% (106/144) | 86.6% (71/82) | 82.2% (60/73) | 0.159 |
| | 6 to 23 months non-breastfed | 45.0% (9/20) | 90.9% (10/11) | 43% (3/7) | 0.922 |
| | 6 to 23 months (all) | 70.1 (141/201) | 89.1 (109/119) | 76.2% (77/101) | 0.265 |
| **Minimal Acceptable Diet** (all 6–23 months old who meet minimum dietary diversity and minimum meal frequency) | | 11.9% (24/201) | 21.0% (25/119) | 30.7% (31/101) | 0.000* |

^Statistical analysis for p-value compares 2020 to 2017 due to varying independence of 2018 sample

* p-value ≤ 0.05 is statistically significant

Minimum meal frequency is the minimum amount of energy intake required by an infant or young child from foods other than breast milk in a day [22]. The minimum meal frequency requirements for an infant or young child in a day are determined by their age and whether or not they are breastfed. Breastfeeding infants aged 6 to 8 months are expected to have at least two meals per day, while breastfed infants aged 9 to 23 months are recommended to have at least three meals per day. Non-breastfed children aged 6 to 23 months are suggested to eat at least four times a day (World Health Organization 2010b). Among all children aged 6–23 months, the percent who met the minimum meal frequency increased from 70% in 2017 to 76% in 2020. The greatest increase in the proportion of children who met the minimum meal frequency was among breastfed children 9 to 23 months old, improving from 74% in 2017 to 82% in 2020.

Minimum acceptable diet (MAD) is a composite indicator combining the dimensions of dietary diversity and meal frequency to assess the proportion of children aged 6–23 months who meet minimum criteria for infant and young child feeding practices. The MAD for breastfed children is achieved when the child consumes at least four out of the standard seven food groups and is fed the recommended minimum number of times or more per day. Non-breastfed children aged 6–23 months are considered to have met the minimum dietary diversity if they have consumed four or more feedings of solid, semi-solid, soft food, or milk. For non-breastfed children to achieve a MAD, they must have at least two milk feedings out of these four feedings.

According to the 2020 report, only 31% of children met the criteria for MAD, which is a statistically significant increase (p<0.001) compared to the baseline where only 12% of children met the criteria.

## Program exposure and change in nutrition and WASH practices among households with children under 5 years of age

Out of the 543 households that had children under 5 years old, 75% reported in 2020 that they had received information on treating diarrhea in children, which was an increase from 59% in the baseline. The majority of the households (84%) received this information from Health Extension Workers (HEWs) in 2020 (Table 7). Among the Growth through Nutrition interventions, the most common sources of information on treating diarrhea in 2020 were ECCs (34% of households) and Saving Group Meetings (30% of households). Almost all households (99%) that received information on treating diarrhea also reported adopting select health or nutritional practices, such as taking their child to a nearby health facility (75% of households in 2020 compared to 58% in the baseline), increasing meal frequency (52% in 2020 compared to 42% in the baseline), and increasing breastfeeding frequency (44% in 2020 compared to 43% in the baseline).

Moreover, in 2020 about 88% of these households also received information on hand washing with soap at critical times, which was an increase from 67% in 2018. HEWs remained the primary source of information on hand washing, identified by 81% of households in 2020, which was consistent with the findings from 2018. Among the project's interventions, women also received information from ECCs (36%) and Saving Group Meetings (28%) in 2020. As a result of this information, 94% of respondents in 2020 reported washing hands with water and soap or ash at crucial times, increasing from 89% in 2018.

## Association of program exposure with key nutrition/WASH practices

Growth through Nutrition focused on implementing social and behavior change communication (SBCC) activities as one of its intervention areas. The goal was to improve awareness of

**Table 7. Sources of information and reported changes in key health, nutrition, WASH, and child feeding practices among households with children under 5 (2017–2020).**

| Did you or anyone in household receive information in the following topics in the past six months? | | 2017 | 2018 | 2020 |
|---|---|---|---|---|
| | | % (N) | % (N) | % (N) |
| **Households with children under 5 years of age residing in the household.** | | **93.8 (362/386)** | **92.4 (330/357)** | **82.5 (543/658)** |
| **Received information on treating your children's diarrhea** | | **58.6 (212/362)** | **37.3 (133/330)** | **75.1 (408/543)** |
| Source | Health Worker | 41.5 (88/212) | 6.8 (9/133) | 38.2 (156/408) |
| | Health Extension Worker | 77.8 (165/212) | 91.7 (122/133) | 83.6 (341/408) |
| | AEWs | 42.9 (91/212) | 33.8 (45/133) | 26.5 (108/408) |
| | HAD/WDA | 12.7 (27/212) | 2.3 (3/133) | 10.3 (42/408) |
| | Enhanced Community Conversation | 0.0 (0/212) | 36.1 (48/133) | 34.1 (139/408) |
| | Cooking Demonstration | 15.1 (32/212) | 1.5 (2/133) | 6.1 (25/408) |
| | MVHHs or Saving Group Meetings | 3.3 (7/212) | 27.8 (37/133) | 30.1 (123/408) |
| | Social/Religious Group Meetings | 0.9 (2/212) | 0.0 (0/133) | 3.4 (14/408) |
| | Radio | 0.9 (2/212) | 1.5 (2/133) | 1.5 (6/408) |
| Practices changed | Increase frequency of BF | 43.4 (85/212) | 27.8 (37/133) | 43.6 (178/408) |
| | Increase Meal Frequency | 42.3 (83/212) | 31.6 (42/133) | 52.2 (213/408) |
| | Rehydrate the Child | 30.6 (60/212) | 10.5 (14/133) | 28.7 (117/408) |
| | Took the child to nearby health facility | 58.2 (114/212) | 37.6 (50/133) | 74.8 (305/408) |
| | Giving ORS and Zinc | 28.1 (55/212) | 32.3 (43/133) | 42.2 (172/408) |
| | Not Practiced any of the above | 15.3 (30/212) | 35.3 (47/133) | 13.7 (56/408) |
| **Received information on hand washing with soap at critical times** | | **78.0 (301/362)** | **66.6 (218/327)** | **87.7 (478/545)** |
| Source | Health Worker | 31.5 (95/301) | 4.6 (10/218) | 36.6 (175/478) |
| | Health Extension Worker | 73.5 (222/301) | 79.8 (174/218) | 80.5 (385/478) |
| | AEWs | 47.4 (143/301) | 30.7(67/218) | 28.0 (134/478) |
| | HDA/WDA | 7.6 (23/301) | 2.3 (5/218) | 11.7 (56/478) |
| | Enhanced Community Conversation | 0.0 (0/301) | 47.7 (104/218) | 36.2 (173/478) |
| | Cooking Demonstration | 13.2 (40/301) | 5.0 (11/218) | 10.7 (51/478) |
| | MVHHs or Saving Group Meetings | 7.6 (23/301) | 17.9 (39/218) | 28.2 (135/478) |
| | Social/Religious Group Meetings | 0.7 (2/301) | 0.0 (0/218) | 3.6 (17/478) |
| | Radio | 0.3 (1/301) | 1.4(3/218) | 1.0 (5/478) |
| | Poster/Flyers/Leaflets | 0.0 (0/301) | 1.4 (3/218) | 6.9 (33/478) |
| Practices changed | Wash hands with water and soap or ash at critical times | 97.0 (293/301) | 89.4 (195/218) | 93.5 (447/478) |
| | Not practiced | 3.0 (9/301) | 10.6 (23/218) | 6.5 (31/478) |

healthy behaviors and create an enabling environment that encouraged adoption of healthy Nutrition and WASH behaviors among MVHH residents. In 2020, 76% of households interviewed (Cohort 1 and Cohort 2) reported participation in Women's Saving Groups, and 83% reported participation in Enhanced Community Conversations (ECC) conducted by Growth through Nutrition in the last six months. The project aimed to evaluate whether participation in these two community-level interventions is associated with improved nutrition and WASH household-level outcomes.

For maternal health, minimum dietary diversity of women and uptake of Iron Folic Acid (IFA) supplements are key outcomes of interest. The 2020 survey data showed that while the proportion of women who met adequate minimum dietary diversity (MDD) improved overall since 2017, participating in ECCs alone was not significantly associated with having adequate MDD, $X^2$ (1, N = 656) = 3.3, p = 0.19. This indicates that increased access to and consumption of diverse food groups resulted from other variables and not participation in ECCs alone. However, there was a significant relationship between participating in ECCs and taking or

purchasing IFA during the last pregnancy, $X^2$ (1, N = 201) = 11.96, p = 0.018. Women who participated in ECCs were more likely to take IFA during pregnancy.

For infant and young child nutrition outcomes, ECC activities aimed to encourage parents to practice exclusive breastfeeding and increase the diversity of foods children consume. The 2020 survey data showed that participating in ECCs was not significantly associated with MDD of children, $X^2$ (1, N = 100) = 1.17, p = 0.558. However, compared to the baseline, there was a significant increase in the proportion of children who had adequate MDD. Similarly, participating in ECCs was not significantly associated with early initiation of breastfeeding under six months, $X^2$ (1, N = 54) = 2.35, p = 0.31, nor with exclusive breastfeeding under six months, $X^2$ (1, N = 55) = 0.08, p = 0.96. The ECCs did not significantly impact breastfeeding practices.

Among Cohort 1 households, there was a significant increase in the proportion of women who had adequate MDD compared to the baseline. One of the project's hypotheses was that supporting households to generate income would increase access to diverse foods, which would, in turn, support women to consume diverse diets during and after pregnancy. Therefore, an increase in household income should impact MDD for women. Respondents from Cohort 1 were categorized into two groups, based on whether or not they reported at least a 10% increase in household income since baseline, to assess if MDD among women was influenced by income change. The data analysis showed that a 10% or more increase in household income alone was not significantly associated with having adequate MDD, $X^2$ (1, N = 307) = 1.18, p = 0.28. Therefore, the significant change in women who had adequate MDD was associated with other factors.

## Food groups consumed by women

The measurement of a mother's dietary diversity is used to assess her access to a variety of foods and the nutritional adequacy of her diet. It is determined by whether or not women aged 15–49 have consumed at least five out of ten defined food groups on the previous day or night [23, 24].

Between 2017 and 2020, the mean number of food groups consumed by women in MVHHs increased slightly from 2.5 food groups in 2017 to 3.4 food groups in 2020. With the exception of dark green vegetables, which decreased from 36% in 2017 to 30% despite being the third most commonly consumed food group in 2020 (Fig 4), consumption of almost all food groups increased among women. Pulses, such as beans, peas, and lentils, saw the highest increase, rising from 56% in 2017 to 76% in 2020. Eggs were the next most increased food group, rising from 1% to 19% over the same period.

Interestingly, despite few children consuming meat or flesh products, 17.2% of households in 2020 reported consuming meat, poultry, or fish products. This may suggest that when households have access to meat, they prioritize its consumption by women or mothers, rather than giving it to children. If this was the case, it could indicate that families understand the importance of dietary diversity, particularly for pregnant or breastfeeding women, and are adopting healthy practices in their homes.

## Minimum dietary diversity–women (MDD-W)

Women of reproductive age (15–49 years) are at a higher risk of suffering from multiple micronutrient deficiencies, which can impact their overall health and ability to perform daily tasks such as caring for their children and participating in income-generating activities [25]. In order to meet the minimum dietary diversity, a woman of reproductive age must consume at least five out of the ten specified food groups within a 24-hour period. Table 8 indicates that

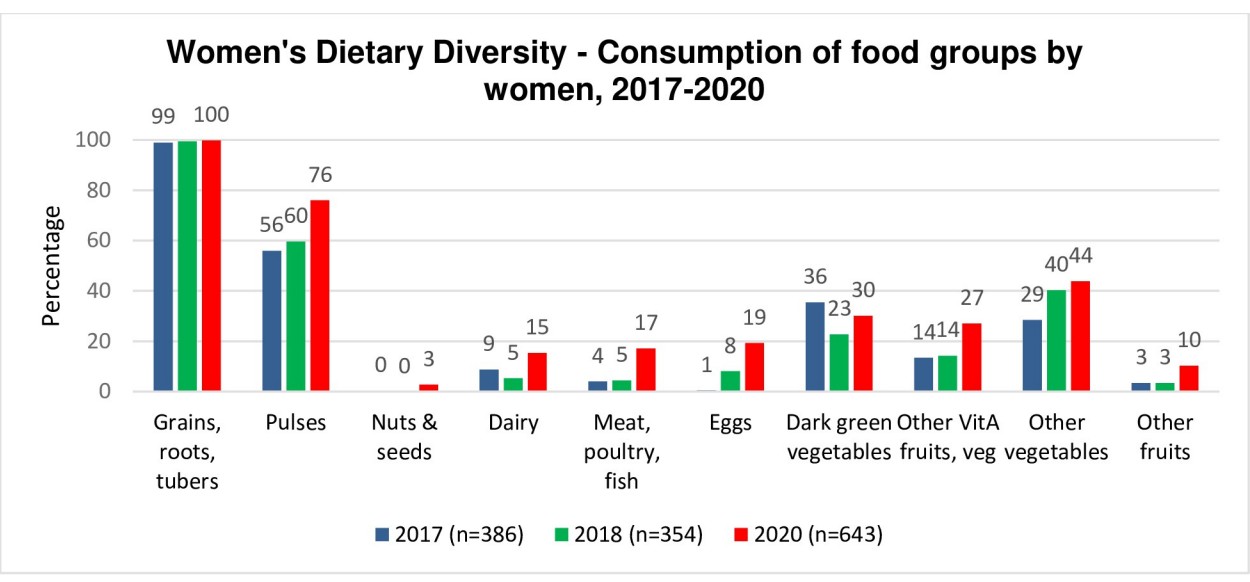

**Fig 4. Consumption of ten food groups by women in MVHHs (2017–2020).**

only 18% of women in the MVHHs consumed five or more food groups in 2020. However, the percentage of women meeting the minimum dietary diversity increased from 2% in 2017 to 16% in 2020, as shown by Cohort 1 data. This increase was statistically significant (p<0.001) and may be attributed to the project's efforts to improve access to diverse foods and promote healthy nutritional practices within households. In fact, 98% of respondents in 2020 credited the project for the positive changes in their diets.

The survey also measured iron folate supplementation among pregnant women. Among Cohort 1, 75% of women who had been pregnant in the last two years received iron folate supplements at least once during their pregnancy, which was an increase from 71% in 2017. However, the proportion of women who received IFA supplementation for at least three months during pregnancy decreased to 28% in 2020, down from 33% in 2017. It is important to note

**Table 8. Dietary diversity and IFA supplementation status among Women in MVHHs, (2017–2020).**

| Women Level Outcomes | | 2017 | 2018 | 2020 (All) | 2020 Cohort 1 only | Level of sig (P-value) |
|---|---|---|---|---|---|---|
| **Mean # food groups consumed from 10 food groups** | | 2.5 groups (386) | 2.6 groups (354) | 3.4 groups (643) | 3.3 groups (307) | |
| **Minimum dietary diversity** (5 food groups or more) | **Inadequate:** 4 food groups or less | 97.6 (377/386) | 91.5 (324/354) | 82.1% (528/643) | 84% (258/307) | .000* |
| | **Adequate:** 5 food groups or more | 2.4% (9/386) | 8.5 (30/354) | 17.9% (115/643) | 16.0% (49/307) | |
| **Proportion of pregnant women who took IFA supplement at least once** (pregnant in last 2 years) | | 70.8% (209/295) | 70.7% (128/181) | 78.1% (157/201) | 74.7% (71/95) | .031* |
| **Proportion of women who received iron and folic acid (IFA) supplementation for at least 3 months during pregnancy** | | 32.5% (68/209) | 40.6% (52/128) | 26.8% (42/157) | 28.2% (20/71) | .116 |
| **Proportion of women who reported the change in dietary diversity was a positive change** | | N/A | 68.3% (243/356) | 79.5% (511/643) | 77.2% (237/307) | .013* (2018 to 2020) |
| **Proportion of women who believe Growth through Nutrition is involved in some way with change in dietary diversity** | | N/A | 96.1% (244/254) | 97.6% (519/532) | 98% (240/245) | .414 (2018 to 2020) |

* p-value ≤ 0.05 is statistically significant

**Table 9. Hand washing practices and access to facilities among MVHHs, (2017–2020).**

| Sanitation and Hygiene | | 2017 | 2018 | 2020 ALL | 2020 Cohort 1 | P-Value |
|---|---|---|---|---|---|---|
| | | % (n) | % (n) | % (n) | % (n) | |
| Yesterday, did you wash your hands | | 99.5 (284/386) | 100 (357/357 | 99.8 (655/656) | 100 (319/319) | |
| If YES, tell all moments you did wash hands | Dirt is visible | 49.5 (190/384) | 56.3 (201/357) | 59.4 (389/655) | 58.6 (187/319) | .026 |
| | *After toilet use/defecation/ urination | 62.2 (239/384) | 70.6 (252/357) | 70.2 (460/655) | 73 (233/319) | .009* |
| | *After cleaning child following defecation | 39.8 (153/384) | 30.0 (107/357) | 42.7 (280/655) | 42.9 (137/319) | .004* |
| | *Before preparing the food | 87.2 (335/384) | 84.0(300/357) | 88.7 (581/655) | 89.3 (285/319) | .078 |
| | Before serving a meal | 55.2 (212/384) | 53.5 (191/357) | 75.4 (494/655) | 76.5 (244/319) | .000* |
| | *Before eating | 87.2 (335/384) | 87.7 (313/357) | 91.3 (598/655) | 91.2 (291/319) | .068 |
| | After eating | 71.6 (275/384) | 61.1 (218/357) | 78.5 (514/655) | 78.7 (251/319) | .000* |
| | *Before feeding a child | 37.2 (143/384) | 44.3 (158/357) | 40.6 (266/655) | 39.8 (127/319) | .682 |
| | When I am reminded to do so | 1.3 (5/384) | 3.9 (14/357) | 11.9 (78/655) | 10.7 (34/319) | .000* |
| *Percent who practice all five critical hand washing moments (highlighted in bold above) | | 17.6 (68/386) | 15.8 (61/357) | 23.1 (151/655) | 23.8 (76/319) | .117 |
| What do you use to wash your hands most of the time? | | | | | | .000* |
| | Water Only | 44.9 (172/384) | 38.9 (139/357) | 40.3 (264/655) | 44.2 (141/319) | |
| | Water and Soap | 52.7 (202/384) | 53.5 (191/357) | 51.6 (338/655) | 46.1 (147/319) | |
| | Water and Ash/Endod | 2.3 (9/384) | 7.6 (27/357) | 8.1 (53/655) | 9.7 (31/319) | |
| Percentage of washing hands with soap /ash/endod and water most of the time | | 55.1 (211/383) | 61.1 (218/357) | 60 (391/655) | 56 (178/319) | .065 |
| Hand washing facility at home | | 2% (8/386) | 14% (51/357) | 25.5 (86/655) | 30.7 (98/319) | .000* |

^Statistical analysis for p-value compares 2020 Cohort 1 only to 2018 and 2017

* p-value ≤ 0.05 is statistically significant

that participation in ECCs was associated with higher IFA uptake. Therefore, pregnant women should be encouraged to attend ECCs regularly to improve their awareness and create a supportive environment that promotes the uptake of IFA supplementation. Additionally, the availability of IFA in health facilities should be improved where needed.

## Sanitation and hygiene

An essential element of the MVHH strategy was providing interpersonal communication and support to promote the adoption of proper sanitation and hygiene practices. Table 9 shows the percentage of household respondents who reported engaging in key behaviors of interest, such as hand washing at critical times during the day and the use of soap or clean water. In the baseline, only 2% of households reported having a hand washing facility at home. However, in 2020, this percentage significantly increased to 26% among all households and 31% among Cohort 1 only. The majority of respondents (76%) indicated that they use a water basin with a jug, and 90% of respondents reported that their hand washing facility was located near the latrine. However, among households with a hand washing facility, only about one-third (36%) had water available at the designated hand washing facility. The increase observed in households with a hand washing facility at home from the baseline to Cohort 1 was statistically significant (p<0.001).

Regarding hand washing, 60% of respondents in 2020 reported washing their hands with soap or ash/endod most of the time, which was a slight increase from baseline (55% in 2017). Almost all respondents, except one (655 out of 656), reported washing their hands the previous day. In 2020, the majority of respondents reported washing their hands before food preparation (89%) and before eating (91%). The least number of respondents reported washing their hands when reminded to do so (12%), before feeding a child (41%), and after cleaning a child

following defecation (43%). Only 23% of respondents in 2020 practiced hand washing at the main critical times (after toilet use/defecation/urination, after cleaning a child following defecation, before preparing food, before eating, and before feeding a child), an improvement from the 18% of households who reported practicing the same behaviors in the baseline. Among Cohort 1 households, the changes in the proportion of households who practiced hand washing after defecation/urination, after cleaning children following defecation, before serving a meal, and after eating were statistically significant. While hand-washing behaviors have generally improved, hand-washing behaviors related to childcare are still lagging behind compared to other time points, indicating the need for additional SBCC and tailored support to households. In addition to improving awareness and encouraging improved nutrition practices, the ECCs also focused on key WASH behaviors to improve child nutrition and health outcomes. Overall, a greater proportion of households reported having hand-washing facilities in 2020 compared to the baseline. However, participating in ECCs was not significantly associated with having hand washing facilities or washing hands before feeding children.

## Limitations of the study

The follow-up survey was conducted during February and March of 2020, whereas the previous surveys were conducted in October and November, which is typically considered a lean harvest season. Therefore, it should be noted that the seasonal effects of the timing of data collection may impact the comparability of data to previous years as households typically experience less food insecurity during the February-March period. Additionally, there may be response bias present in the data from Cohort 1 as some respondents may have been able to anticipate the questions from previous years and provide answers they believe data collectors wanted to hear. Moreover, recall bias may also have influenced data pertaining to the previous day and year.

It is important to note that the number of households from Cohort 1 with recently pregnant women and children under 2 was significantly lower in 2020 compared to the baseline. Therefore, comparisons between the 2020 findings and the baseline should be interpreted cautiously due to the unequal variance between samples, resulting in a loss of statistical power. Furthermore, since children from Cohort 1 aged out of the cohort, the findings for Infant and Young Child Feeding (IYCF) indicators were mainly from the new cohort of households, named Cohort 2. Therefore, any differences in the 2020 data and previous years' data cannot be interpreted as improvements in the same households.

## Discussion and recommendations

### Discussion

The study aimed to systematically generate evidence on project implementation and monitor changes in nutrition and WASH behaviors in economically vulnerable households to improve nutrition and health for children, pregnant, and lactating women, with a focus on the first 1,000 days of life. Regarding project targeted changes in feeding practices among mothers and chidren in MVHHs, the assessment has shown a significant change in key indicators over the years. Child dietary diversity significantly increased (P<0.001) since the baseline, alongside the significant increase in consumption of eggs, dairy products, and Vitamin A foods, which was a key part of the livelihood and SBCC support provided to MVHHs. The livelihood and SBCC supports to MVHHs are effectively reaching and improving household practices, and therefore are likely to have an impact on nutrition outcomes. Notably, while minimum dietary diversity improved, the proportion of households reporting early initiation of breastfeeding and exclusive breastfeeding marginally decreased since the baseline. Therefore, there is a need to strengthen antenatal nutrition counseling to improve early initiation and exclusive breastfeeding mainly

through Health Extension Workers and Health workers which are identified as major sources of information. Besides, feeding practices revealed some decline from 2017 to 2018, followed by a subsequent increase from 2018 to 2020. This fluctuation deserves further retrospective exploration to ascertain the events that transpired in the study areas during this timeframe.

Similarly, the project has succeeded in bringing about desired changes in women's nutrition in MVHHs. Minimum Dietary Diversity of women significantly increased (P<0.001) since the baseline due to increased consumption of dairy, eggs, vegetables, meat, and pulses. Increasing the variety of foods in the diet helps to ensure adequate intake of essential nutrients, one of the most important approaches responsible for addressing maternal micronutrient deficiency [23, 24, 26]. The proportion of pregnant women who received iron folate supplementation at least once improved, but the proportion that received supplementation for at least 3 months during pregnancy decreased slightly. Such inadequate duration of IFA uptake coupled with declining trends observed in dietary intake of dark green vegetables among mothers implies the need to ensure women receive continued support during pregnancy, both in terms of addressing behavior and supply-related issues.

Besides, the project's household-level WASH interventions aimed to reduce the prevalence of diarrheal disease and potential effects of environmental enteropathy among children, a cause of childhood undernutrition [27]. The study found positive improvements in the proportion of households that reported having a handwashing facility at home and washing hands with water and soap or ash/endod. Remarkably, there was a greater change in the proportion of households that wash with Endod specifically, indicating some local preference over the use of soap. Endod plants have been used as a detergent and as traditional medicine for centuries in Ethiopia [28]. Hand washing is an effective way to prevent infections as a number of infectious diseases can be spread from one person to another by contaminated hands [29]. The availability of water at handwashing facilities was low, highlighting the need to improve availability of water and soaps among households to create a hygiene friendly environment to support the uptake of handwashing behaviors. Here, It is worth noting that the lack of a significant association between participation in ECC sessions and having handwashing facilities, or washing hands before feeding children, should be explored further. This investigation should consider whether the issue is related to resource needs beyond awareness, the approach of the behavior change interventions, and/or the time-consuming nature of behavior change communications to effect desired changes.

In cases of both optimal nutrition and WASH practices, the results of the 2020 survey showed an overall improvement in nutrition, livelihood, and WASH practices in MVHHs compared to the baseline. To determine the impact of the project's activities on these changes, an analysis was conducted to identify if exposure to ECCs, Saving Groups, or direct technical support visits were associated with the observed improvements. The findings indicated that only IFA uptake among pregnant women was significantly associated with participation in ECC activities, while there were no significant associations for other indicators such as MDD, exclusive breastfeeding, and raising chicken.

However, the statistically significant changes in the proportion of households practicing select behaviors compared to 2017 suggest that the cumulative support provided to MVHHs was encouraging behavior change. The analysis also indicates that exposure to multiple activities is more likely to result in behavior change rather than any one activity alone, such as participation in ECCs or Savings Groups.

Overall, the findings suggest that the project's activities positively influenced nutrition and WASH related behavior change in MVHHs, but further data analysis, along with comparison to a control group, is required to assess the effect of the project's activities on observed improvements in health behaviors or outcomes.

## Recommendations

The study indicated an improvement in dietary diversity for women and children. However, it remains concerning that a significant percentage of households still lack adequate minimum dietary diversity. This is evident from the low consumption levels of crucial food groups among the household's most nutritionally vulnerable members. To improve dietary diversity, there is a need to strengthen evidence-based project activities, not limited to SBCC activities, that increase access to and consumption of nutritious foods among the most vulnerable households.

The observed increase in the proportion of women receiving iron and folic acid (IFA) supplementation during pregnancy indicates a dual need. Firstly, there is a necessity to enhance support for quality antenatal care counseling and social and behavior change communication (SBCC) activities focused on maternal nutrition. Secondly, to meet the growing community demand, it is crucial to address the accessibility challenges related to IFA resources. Despite reports of improved overall hand-washing practices, significant gaps persist, particularly at critical times such as after cleaning children following defecation or toilet use, and before feeding children. These shortcomings pose serious health risks to children due to inadequate hand-washing. This highlights the possibility that participation in SBCC sessions alone may not sufficiently drive positive changes in WASH-related practices. Hence, it is imperative to not only focus on enhancing SBCC activity messaging but also to prioritize enhancing the availability and accessibility of water for community use, as this will play a pivotal role in driving meaningful improvements.

Furthermore, the proximity of water facilities to latrines makes it inconvenient for individuals to access them before feeding or caring for their children, which is when hand washing is least practiced. Therefore, households require additional support to improve the availability and accessibility of water and soap to make hand washing easier when needed.

The study also revealed that SBCC platforms could better utilize certain sources of information to reach households. Health extension workers (HEWs) were the most common source of information on maternal, newborn, and child health (MNCH) and nutrition issues, making them an essential cadre to continue supporting. Although households have access to diverse channels of information, including HEWs, health providers, Savings Groups, and AEWs, fewer respondents identified AEWs as a source of health information. Therefore, similar integrated projects should focus on the role of AEWs in supporting agronomic practices instead and support HEWs as an additional health resource to strengthen more commonly used platforms to deliver health and nutrition SBCC to MVHHs. Similarly, given the declining trend in the effectiveness of project-specific and time-bound Social and Behavior Change Communication (SBCC) interventions as these projects conclude, it is essential to strengthen routine sources of information through the health sector should be prioritized as a sustainable source to the community.

## Supporting information

**S1 Data.**
(XLSX)

## Acknowledgments

The survey team would like to express its gratitude to the Woreda Health, Agriculture, and Administration Offices for their unwavering support and permission to conduct the survey in each woreda. We would also like to extend our sincere appreciation to all participants who

graciously responded to the lengthy interviews. Without their kind participation, this study would not have been possible. The Growth through Nutrition team deserves special thanks for their exceptional leadership and coordination of the fieldwork in each region. Additionally, the authors would like to acknowledge Dr. Yigzaw Desalegn for providing valuable feedback on the development of study tools and Dr. Larry Dershem for his substantial contribution to the study's design. The authors gratefully acknowledge the contributions of Abise Gudeta and Behailu Wodegiorgis in the development of study tools and their invaluable support in data collection.

## Author Contributions

**Conceptualization:** Daniel Abbott.

**Data curation:** Adam Bailes.

**Formal analysis:** Cherinet Abuye, Adam Bailes.

**Investigation:** Daniel Abbott.

**Methodology:** Cherinet Abuye.

**Project administration:** Cherinet Abuye, Daniel Abbott, Lioul Berhanu, Rachel Holtzman.

**Resources:** Daniel Abbott.

**Supervision:** Cherinet Abuye, Daniel Abbott, Lioul Berhanu, Rachel Holtzman.

**Writing – original draft:** Cherinet Abuye, Daniel Abbott, Adam Bailes.

**Writing – review & editing:** Cherinet Abuye, Lioul Berhanu, Adam Bailes, Rachel Holtzman.

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
