## [Decision Letter · Decision Letter 0]

20 Mar 2024

PONE-D-23-39220Nutrition and water, sanitation and hygiene (WASH) practices and changes among most vulnerable households (MVHHs) benefiting from growth through nutrition project interventions in Ethiopia.PLOS ONE

Dear Dr. Tirore,

Thank you for submitting your manuscript to PLOS ONE. After careful consideration, we feel that it has merit but does not fully meet PLOS ONE’s publication criteria as it currently stands. Therefore, we invite you to submit a revised version of the manuscript that addresses the points raised during the review process.

We look forward to receiving your revised manuscript.

Kind regards,

Muhammad Ali, PhD

Academic Editor

PLOS ONE

Additional Editor Comments:

Dear Authors

Thank you for submitting your paper to Plos One. We have received two reviews on your paper. Please incorporate their comments in your revision. In addition, I also have the following comments on your manuscript that require your attention:

1) Your title explicitly suggests that your paper discusses WASH and nutrition, however, your abstract only mentions handwashing briefly. I expected more results related to Water, Sanitation and Hygiene. I strongly recommend changing the title to reflect the object of the study. It took me some time to understand that this paper is basically an impact evaluation of a project that aims to improve both WASH access and nutrition outcomes. Also revise the main section headings. these headings are too long and confusing eg page 10 and 12.

2) Objective of the study on page 4 is unclear because it has two "ands". Please rewrite it for clarity.

3) Introduction of the paper is too short. There is hardly any discussion on the state of the art except for the first paragraph. why is it necessary to improve WASH access or feeding practices? discuss it please with the help of existing literature.

4) Table 1 suggests that the sample used in this study mostly consists of ever-married women. However, this specification of the sample is not mentioned in the title. It should be clearly mentioned that the study is on women and children.

5) It Table 4, feeding practices are mostly decreasing between 2017 and 2018 substantially and then increasing between 2018 and 2020 again. These indicators are usually not so volatile. What is happening there?

6) What is the difference between heading on Page 10 and Page 12? children aged 6 -23 months are also under 2 years old. And in this paper, nutrition seems to be used synonymous with feeding. if that is the case, both headings are identical.

7) Please rethink the arrangement of arguments. it is confusing right now. Group similar topics together and attempt to establish a transition and relationship among the arguments. If you want to organize your discussion by food and WASH then clearly number the headings and discuss these two domains under their respective headings. When age groups change, give separate sections to ages and number them properly. It will help in understanding the thought process.

8) In the discussion, talk about the objectives of the project and one by one explain what it managed to achieve and what it did not. Otherwise arguments are lost in words.

All the best.

Reviewers' comments:

Reviewer's Responses to Questions

**Comments to the Author**

1. Is the manuscript technically sound, and do the data support the conclusions?

Reviewer #1: Yes

Reviewer #2: Yes

2. Has the statistical analysis been performed appropriately and rigorously? 

Reviewer #1: No

Reviewer #2: Yes

3. Have the authors made all data underlying the findings in their manuscript fully available?

Reviewer #1: No

Reviewer #2: Yes

4. Is the manuscript presented in an intelligible fashion and written in standard English?

Reviewer #1: Yes

Reviewer #2: Yes

5. Review Comments to the Author

Reviewer #1: This is a good study, however, the analysis could have included other dimensions without involving advanced econometric techniques. For instance, it would be interesting to explore how WASH practices and nutritional intake of the infants change with education of the mother The authors are suggested to add two-way and three-way cross tabulations.

Reviewer #2: Excellent study. But please add a section of behavior change models as adoption of WASH practices is a long run behaviour change. For instance, see Ginja, S., Gallagher, S., & Keenan, M. (2021). Water, sanitation and hygiene (WASH) behaviour change research: why an analysis of contingencies of reinforcement is needed. International Journal of Environmental Health Research, 31(6), 715-728.

6. PLOS authors have the option to publish the peer review history of their article (what does this mean?). If published, this will include your full peer review and any attached files.

Reviewer #1: No

Reviewer #2: **Yes: **Dr. Ayesha Nazuk

---

## [Author Response · Author response to Decision Letter 0]

12 Jun 2024

If applicable, we recommend that you deposit your laboratory protocols in protocols. …

 Author’s response: Not applicable 

https://journals.plos.org/plosone/s/file?id=wjVg/PLOSOne_formatting_sample_main_body.pdf ...

a. Author’s response: The manuscript was prepared following PLOS ONE's style requirements. The files are named as "Revised Manuscript" with track changes, "Manuscript" without track changes and main article file.

a. Authors Response: The correct grant number is “USAID Cooperative Agreement No. AID 663-A-16-00007.” The authors used this funding to conduct the study.

3. If there are ethical or legal restrictions on sharing a de-identified data set, please explain them in detail …

a. Author’s response: There are no specific ethical/legal restrictions in accessing the dataset. All relevant data are within the manuscript and its Supporting Information files. The data is provided in an Excel file named "Final Merged Data Three Files_MVHH_Mar_14_2020."

4. Your ethics statement should only appear in the Methods section of your manuscript. If your ethics statement is written in any section besides the Methods, …

a. Author’s response: Ethics statement is written only in methodology - in the manuscript.

Additional editor Comments:

1. Your title explicitly suggests that your paper discusses WASH and nutrition, however, your abstract only mentions hand washing briefly. I expected more results related to Water, Sanitation and Hygiene …

a. Author’s response: The abstract now incorporates concise findings on Water, Sanitation, and Hygiene in addition to the reported hand washing practice. The title has been adjusted to better align with the study's objective. Besides, the headings on pages 10 and 12 have been condensed for clarity.

2. Objective of the study on page 4 is unclear because it has two "ands". Please rewrite it for clarity. 

a. Author’s response: Corrected

3. Introduction of the paper is too short. …

a. Author’s response: This comment is addressed in the introduction section by inserting additional evidence meant to illustrate why access to WASH (Water, Sanitation, and Hygiene) is necessary.

4. Table 1 suggests that the sample used in this study mostly consists of ever-married women. However, this specification of the sample is not mentioned …

a. Author’s response: The Study title refers to ‘Households” not specifically to ‘women and children as, the sampling is not limited to women and children but caregivers in the households who can respond for the questionnaires. The data included male (e.g. 6.6% in 2020) as shown in this table.

5. In Table 4, feeding practices are mostly decreasing between 2017 and 2018 substantially and then increasing between 2018 and 2020 again. These indicators are usually not so volatile. …

a. Author’s response: This was explained in the discussion section, highlighting the necessity for additional exploration to determine the events that occurred in the study areas during this timeframe

6. What is the difference between heading on Page 10 and Page 12? children aged 6 -23 months are also under 2 years old. And in this paper, nutrition seems to be used synonymous with feeding. if that is the case, both headings are identical.

a. Author’s response: Shifted the order of the sections with titles modified so that the narrative flows from ‘breastfeeding’ to ‘complementary feeding’/ Presented in two separate sections for the sake of clarity/ for better flow.

7. Please rethink the arrangement of arguments. it is confusing right now. Group similar topics together and attempt to establish a transition and relationship among the arguments. …

a. Author’s response: rearranged the presentation in the Discussion section by starting with nutrition/feeding related issues among mothers and children and proceeding to WASH. This is in line with the revised order in the Results section - that goes from pregnancy, breastfeeding, complementary feeding, maternal nutrition to WASH practices. Similar rearrangement is done for the recommendation section. While headings clearly reflect the flow in the Results section, separate sections are not used in the Discussion and Recommendation sections.

8. In the discussion, talk about the objectives of the project and one by one explain what it managed to achieve and what it did not. Otherwise arguments are lost in words.

a. Author’s response: Addressed together with the above comment by adding/modifying the objectives that the project aimed to fulfil and reporting related changes at each paragraph in the Discussion section.

Reviewers' comments:

Reviewer's Responses to Questions:

Reviewer's Responses to Questions:

1. Is the manuscript technically sound, and do the data support the conclusions? …

Reviewer #1: Yes

Reviewer #2: Yes

Author’s response: Ok

2. Has the statistical analysis been performed appropriately and rigorously?

Reviewer #1: No, 

Author’s response to reviewer #1: We believe that the statistical analysis has been conducted properly and carefully, aligning with the objectives of the study. The objective was to evaluate changes associated with interventions aimed at Most Vulnerable Households within the Growth through Nutrition Project. Given that this is the primary aim of the study, we presume that the statistical analysis has been conducted appropriately and rigorously.

3. Have the authors made all data underlying the findings in their manuscript fully available?

Author’s response to reviewer #1: Raw data will be made available to public via https://data.usaid.gov/browse?category=Health

4. Is the manuscript presented in an intelligible fashion and written in standard English?

Reviewer #1: Yes

Reviewer #2: Yes

Author’s response: Ok

5. Review Comments to the Author

Reviewer #1: This is a good study, however, the analysis could have included other dimensions without involving advanced econometric techniques. For instance, it would be interesting to explore how WASH practices and nutritional intake of the infants change with education of the mother. The authors are suggested to add two-way and three-way cross tabulations.

Author’s response: This is a good recommendation. However, as the primary objective of the study was limited to assessing changes over time among project supported households and changes associated with project-specific intervention (e.g. behavior change sessions) than seeking for association/correlation between other possible underlying factors. 

Reviewer #2: Excellent study. But please add a section of behavior change models as adoption of WASH practices is a long run behaviour change. For instance, see Ginja, S., Gallagher, S., & Keenan, M. (2021). Water, sanitation and hygiene (WASH) behaviour change research: why an analysis of contingencies of reinforcement is needed. International Journal of Environmental Health Research, 31(6), 715-728.

Author’s response: Enriching feedback.

Our reservation is as SBCC sessions complement material support and technical assistance to MVHHs, a focus on BCC model might shift the analysis of the study more to the BCC component. As this study is not originally planned with the explanatory framework presented in the article- analysis of contingencies of reinforcement- in mind, this makes discussion based on such concept less robust.

---

## [Editor Report · Decision Letter 1]

14 Jun 2024

PONE-D-23-39220R1An evaluation of interventions within a growth through nutrition project aimed at enhancing both WASH and nutrition practices among most vulnerable households (MVHHs) in Ethiopia

PLOS ONE

Dear Dr. Tirore,

Thank you for submitting your manuscript to PLOS ONE. After careful consideration, we feel that it has merit but does not fully meet PLOS ONE’s publication criteria as it currently stands. Therefore, we invite you to submit a revised version of the manuscript that addresses the points raised during the review process.

**Academic Editor's Comments**

In title authors mention Growth Through Nutrition in smaller case. I think it is the title of the project that has been evaluated. The title of the projects should be mentioned with all first letters capital to ensure that the readers do not confuse it with simple text.

Please write abstract in one paragraph and attempt to make it a little bit more concise.

Page 1 line 20: Two cohorts in 2017, 2018 and 2020 is confusing? On Page 7 lines 151 and 152, authors mention “both dependent and independent of the baseline sample”. It is unclear what authors mean by it. Please explain. Also, some tables have information about Cohorts in 2020, others don’t. Please make it consistent.

Page 2 Line 31, thank you for adding some information about water and sanitation as well. However, it seems as if the study was primarily focused on handwashing and there is not much information about water and sanitation. It is completely fine if this is the case. Authors can then avoid using the term WASH and mention handwashing instead. If water and sanitation were indeed analyzed then I suggest to define the type of water variables and sanitation variables studied and their corresponding findings.

Page 4 lines 92-94: please replace this part with a statement of what each broad section covers in the rest of the paper.

Please merge the objectives inside the introduction section instead of giving a separate heading for it.

I apologize for not noticing it earlier but there is hardly any discussion on previous studies on this topic. Only first two paragraphs of introduction appear to mention a couple of them. It can be improved in two ways: 1) add a literature review section covering the previous studies on this topic, or 2) add these studies in the introduction section to develop the motivation, not only for this study but also for the project. It is quite possible that the project documents have mentioned these studies. 

Please mention sources of the tables and figures at their bottom. 

In the recommendations section, I suggest that authors should avoid stating the results. If these results are not mentioned in the results section, I suggest to move them to the results section otherwise simple suggest the key recommendations in this section. It is, in my opinion, too lengthy at the moment because it is stating results.

We look forward to receiving your revised manuscript.

Kind regards,

Muhammad Ali, PhD

Academic Editor

PLOS ONE

Journal Requirements:

Additional Editor Comments:

Dear Authors,

I am grateful for your detailed responses and thorough incorporation of our comments. I believe that your paper is very close to a nice publishable shape. I request you to please incorporate the following comments in your next revision.

In title authors mention Growth Through Nutrition in smaller case. I think it is the title of the project that has been evaluated. The title of the project should be mentioned with all first letters capital to ensure that the readers do not confuse it with simple text.

Please write abstract in one paragraph and attempt to make it a little bit more concise.

Page 1 line 20: Two cohorts in 2017, 2018 and 2020 is confusing? On Page 7 lines 151 and 152, authors mention “both dependent and independent of the baseline sample”. It is unclear what authors mean by it. Please explain. Also, some tables have information about Cohorts in 2020, others don’t. Please make it consistent.

Page 2 Line 31, thank you for adding some information about water and sanitation as well. However, it seems as if the study was primarily focused on handwashing and there is not much information about water and sanitation. It is completely fine if this is the case. Authors can then avoid using the term WASH and mention handwashing instead. If water and sanitation were indeed analyzed then I suggest to define the type of water variables and sanitation variables studied and their corresponding findings. There are some findings mentioned in the text for sanitation but the paper is almost silent on water access which is the "W" of WASH.

Page 4 lines 92-94: please replace this part with a statement of what each broad section covers in the rest of the paper.

Please merge the objectives inside the introduction section instead of giving a separate heading for it.

I apologize for not noticing it earlier but there is hardly any discussion on previous studies on this topic. Only first two paragraphs of introduction appear to mention a couple of them. It can be improved in two ways: 1) add a literature review section covering the previous studies on this topic, or 2) add these studies in the introduction section to develop the motivation, not only for this study but also for the project. It is quite possible that the project documents have mentioned these studies. You can refer to them for guidance.

Please mention sources of the tables and figures at their bottom.

In the recommendations section, I suggest that authors should avoid stating the results again. If these results are not mentioned in the results section, I suggest to move them to the results section otherwise simply suggest the key recommendations in this section. It is, in my opinion, too lengthy at the moment because of the results.

Thank you

Muhammad Ali

---

## [Author Response · Author response to Decision Letter 1]

19 Jul 2024

Academic Editor's Comments

Comment 1: In title authors mention Growth Through Nutrition in smaller case. I think it is the title of the project that has been evaluated. The title of the projects should be mentioned with all first letters capital to ensure that the readers do not confuse it with simple text. 

Authors’ response: The title has been adjusted to address reviewers' feedback on capitalization. While one reviewer suggested narrowing the focus to hand-washing practices, we retained the broader term "WASH" (Water, Sanitation, and Hygiene). This is because the study’s findings encompass not only hand-washing and hygiene practices but also the construction and household-level use of latrine facilities. However, the project’s efforts to increase access to safe water were targeted at the community level rather than specifically at Most Vulnerable Households (MVHHs). Consequently, the direct impact of improved water access is not analyzed here.

Comment 2: Please write abstract in one paragraph and attempt to make it a little bit more concise. 

Authors’ response: The abstract has been condensed into a single, cohesive paragraph. This reorganized version briefly summarizes the study's scope, findings, and conclusions, enhancing clarity and focus for the readers.

Comment 3: Page 1 line 20: Two cohorts in 2017, 2018 and 2020 is confusing? On Page 7 lines 151 and 152, authors mention “both dependent and independent of the baseline sample”. It is unclear what authors mean by it. Please explain. Also, some tables have information about Cohorts in 2020, others don’t. Please make it consistent.

Authors’ response: This was revised in the manuscript to address the reviewer's concerns and prevent any potential confusion. 

Comment 4: Page 2 Line 31, thank you for adding some information about water and sanitation as well. However, it seems as if the study was primarily focused on handwashing and there is not much information about water and sanitation. It is completely fine if this is the case. Authors can then avoid using the term WASH and mention handwashing instead. If water and sanitation were indeed analyzed then I suggest to define the type of water variables and sanitation variables studied and their corresponding findings.

Authors’ response: As it has been indicated in the title section, WASH is still kept (despite one reviewers’ suggestion to narrow title to handwashing practice) as the findings presented are not limited to handwashing/hygiene but also refer to construction of latrine facilities and household-level use of latrine facilities. However, activities of the project towards increasing access to safe water was targeted at community level, not specific to MVHHs, thus the direct contribution of increased access to water is not analyzed here. 

Comment 5: Page 4 lines 92-94: please replace this part with a statement of what each broad section covers in the rest of the paper.

Authors’ response: Corrections have been made as recommended by the reviewers.

Comment 6: Please merge the objectives inside the introduction section instead of giving a separate heading for it.

Authors’ response: The study's objectives were integrated into the introduction section.

Comment 7: I apologize for not noticing it earlier but there is hardly any discussion on previous studies on this topic. Only first two paragraphs of introduction appear to mention a couple of them. It can be improved in two ways: 1) add a literature review section covering the previous studies on this topic, or 2) add these studies in the introduction section to develop the motivation, not only for this study but also for the project. It is quite possible that the project documents have mentioned these studies.

Authors’ response: The revised introduction now includes a comprehensive global literature review. As a result of adding new references and modifying existing ones, adjustments were made to the references in sections outside the Introduction to maintain consistency and accuracy throughout the document.

Comment 8: Please mention sources of the tables and figures at their bottom. 

Authors’ response: The tables and figures in this manuscript are derived directly from the raw data generated during the study. They are original and have not been sourced or quoted from any external references.

Comment 8: In the recommendations section, I suggest that authors should avoid stating the results. If these results are not mentioned in the results section, I suggest to move them to the results section otherwise simple suggest the key recommendations in this section. It is, in my opinion, too lengthy at the moment because it is stating results.

Authors’ response: It is revised to limit the inclusion of findings/ results in the recommendation section, as per the reviewer’s feedback.

---

## [Editor Report · Decision Letter 2]

23 Jul 2024

PONE-D-23-39220R2An evaluation of interventions within a Growth Through Nutrition project aimed at enhancing optimal nutrition and water, sanitation and hygiene (WASH) and nutrition practices among nutritionally most vulnerable households (MVHHs) in EthiopiaPLOS ONE

Dear Dr. Tirore,

Thank you for submitting your manuscript to PLOS ONE. After careful consideration, we feel that it has merit but does not fully meet PLOS ONE’s publication criteria as it currently stands. Therefore, we invite you to submit a revised version of the manuscript that addresses the points raised during the review process.

We look forward to receiving your revised manuscript.

Kind regards,

Muhammad Ali, PhD

Academic Editor

PLOS ONE

Journal Requirements:

Additional Editor Comments:

Dear editors,

Thank you for incorporating our comments. I just have one final request. I agree with your justification to use WASH in the paper. Can you please add a sentence in the introduction to show that water interventions were a part of the program but they were not specific to the selected group.

Thank you

---

## [Author Response · Author response to Decision Letter 2]

23 Jul 2024

Academic Editor's Comments

Comment 1: In title authors mention Growth Through Nutrition in smaller case. I think it is the title of the project that has been evaluated. The title of the projects should be mentioned with all first letters capital to ensure that the readers do not confuse it with simple text. 

Authors’ response: The title has been adjusted to address reviewers' feedback on capitalization. While one reviewer suggested narrowing the focus to hand-washing practices, we retained the broader term "WASH" (Water, Sanitation, and Hygiene). This is because the study’s findings encompass not only hand-washing and hygiene practices but also the construction and household-level use of latrine facilities. However, the project’s efforts to increase access to safe water were targeted at the community level rather than specifically at Most Vulnerable Households (MVHHs). Consequently, the direct impact of improved water access is not analyzed here.

Comment 2: Please write abstract in one paragraph and attempt to make it a little bit more concise. 

Authors’ response: The abstract has been condensed into a single, cohesive paragraph. This reorganized version briefly summarizes the study's scope, findings, and conclusions, enhancing clarity and focus for the readers.

Comment 3: Page 1 line 20: Two cohorts in 2017, 2018 and 2020 is confusing? On Page 7 lines 151 and 152, authors mention “both dependent and independent of the baseline sample”. It is unclear what authors mean by it. Please explain. Also, some tables have information about Cohorts in 2020, others don’t. Please make it consistent.

Authors’ response: This was revised in the manuscript to address the reviewer's concerns and prevent any potential confusion. 

Comment 4: Page 2 Line 31, thank you for adding some information about water and sanitation as well. However, it seems as if the study was primarily focused on handwashing and there is not much information about water and sanitation. It is completely fine if this is the case. Authors can then avoid using the term WASH and mention handwashing instead. If water and sanitation were indeed analyzed then I suggest to define the type of water variables and sanitation variables studied and their corresponding findings.

Authors’ response: As it has been indicated in the title section, WASH is still kept (despite one reviewers’ suggestion to narrow title to handwashing practice) as the findings presented are not limited to handwashing/hygiene but also refer to construction of latrine facilities and household-level use of latrine facilities. However, activities of the project towards increasing access to safe water was targeted at community level, not specific to MVHHs, thus the direct contribution of increased access to water is not analyzed here. 

Comment 5: Page 4 lines 92-94: please replace this part with a statement of what each broad section covers in the rest of the paper.

Authors’ response: Corrections have been made as recommended by the reviewers.

Comment 6: Please merge the objectives inside the introduction section instead of giving a separate heading for it.

Authors’ response: The study's objectives were integrated into the introduction section.

Comment 7: I apologize for not noticing it earlier but there is hardly any discussion on previous studies on this topic. Only first two paragraphs of introduction appear to mention a couple of them. It can be improved in two ways: 1) add a literature review section covering the previous studies on this topic, or 2) add these studies in the introduction section to develop the motivation, not only for this study but also for the project. It is quite possible that the project documents have mentioned these studies.

Authors’ response: The revised introduction now includes a comprehensive global literature review. As a result of adding new references and modifying existing ones, adjustments were made to the references in sections outside the Introduction to maintain consistency and accuracy throughout the document.

Comment 8: Please mention sources of the tables and figures at their bottom. 

Authors’ response: The tables and figures in this manuscript are derived directly from the raw data generated during the study. They are original and have not been sourced or quoted from any external references.

Comment 8: In the recommendations section, I suggest that authors should avoid stating the results. If these results are not mentioned in the results section, I suggest to move them to the results section otherwise simple suggest the key recommendations in this section. It is, in my opinion, too lengthy at the moment because it is stating results.

Authors’ response: It is revised to limit the inclusion of findings/ results in the recommendation section, as per the reviewer’s feedback.

---

## [Editor Report · Decision Letter 3]

6 Aug 2024

PONE-D-23-39220R3An evaluation of interventions within a Growth Through Nutrition project aimed at enhancing optimal nutrition and water, sanitation and hygiene (WASH) and nutrition practices among nutritionally most vulnerable households (MVHHs) in EthiopiaPLOS ONE

Dear Dr. Tirore,

Thank you for submitting your manuscript to PLOS ONE. After careful consideration, we feel that it has merit but does not fully meet PLOS ONE’s publication criteria as it currently stands. Therefore, we invite you to submit a revised version of the manuscript that addresses the points raised during the review process.

We look forward to receiving your revised manuscript.

Kind regards,

Muhammad Ali, PhD

Academic Editor

PLOS ONE

Journal Requirements:

Additional Editor Comments:

**Dear Dr. Abuye, Please add the statement about community level water interventions in the paper and upload the manuscript again. Everything else is fine.**

Warm regards

Ali

---

## [Author Response · Author response to Decision Letter 3]

10 Aug 2024

Additional Editor Comments: Please add the statement about community level water interventions in the paper and upload the manuscript again.

Author’s response: Regarding the comment on community-level water interventions, we crafted the following paragraph and included it in the introduction section of the manuscript: "Efforts to increase access to safe water were targeted at the community level, while household interventions focused on promoting hygienic practices such as handwashing and proper latrine construction and use. Although water supply schemes were included in the intervention, they were not exclusive to the population under study. Consequently, we did not analyze the households' water access." 

Journal Requirements: Please review your reference list to ensure that it is complete and correct. If you have cited papers that have been retracted, please include the rationale for doing so in the manuscript text, or remove these references and replace them with relevant current references. Any changes to the reference list should be mentioned in the rebuttal letter that accompanies your revised manuscript. If you need to cite a retracted article, indicate the article’s retracted status in the References list and also include a citation and full reference for the retraction notice.

Author’s response: Due to the addition of new references and modifications to existing ones, adjustments were made to the references in sections beyond the Introduction to ensure consistency and accuracy throughout the document.

Previous Academic Editor's Comments

Comment 1: In title authors mention Growth Through Nutrition in smaller case. I think it is the title of the project that has been evaluated. The title of the projects should be mentioned with all first letters capital to ensure that the readers do not confuse it with simple text. 

Authors’ response: The title has been adjusted to address reviewers' feedback on capitalization. While one reviewer suggested narrowing the focus to hand-washing practices, we retained the broader term "WASH" (Water, Sanitation, and Hygiene). This is because the study’s findings encompass not only hand-washing and hygiene practices but also the construction and household-level use of latrine facilities. However, the project’s efforts to increase access to safe water were targeted at the community level rather than specifically at Most Vulnerable Households (MVHHs). Consequently, the direct impact of improved water access is not analyzed here.

Comment 2: Please write abstract in one paragraph and attempt to make it a little bit more concise. 

Authors’ response: The abstract has been condensed into a single, cohesive paragraph. This reorganized version briefly summarizes the study's scope, findings, and conclusions, enhancing clarity and focus for the readers.

Comment 3: Page 1 line 20: Two cohorts in 2017, 2018 and 2020 is confusing? On Page 7 lines 151 and 152, authors mention “both dependent and independent of the baseline sample”. It is unclear what authors mean by it. Please explain. Also, some tables have information about Cohorts in 2020, others don’t. Please make it consistent.

Authors’ response: This was revised in the manuscript to address the reviewer's concerns and prevent any potential confusion. 

Comment 4: Page 2 Line 31, thank you for adding some information about water and sanitation as well. However, it seems as if the study was primarily focused on handwashing and there is not much information about water and sanitation. It is completely fine if this is the case. Authors can then avoid using the term WASH and mention handwashing instead. If water and sanitation were indeed analyzed then I suggest to define the type of water variables and sanitation variables studied and their corresponding findings.

Authors’ response: As it has been indicated in the title section, WASH is still kept (despite one reviewers’ suggestion to narrow title to handwashing practice) as the findings presented are not limited to handwashing/hygiene but also refer to construction of latrine facilities and household-level use of latrine facilities. However, activities of the project towards increasing access to safe water was targeted at community level, not specific to MVHHs, thus the direct contribution of increased access to water is not analyzed here. 

Comment 5: Page 4 lines 92-94: please replace this part with a statement of what each broad section covers in the rest of the paper.

Authors’ response: Corrections have been made as recommended by the reviewers.

Comment 6: Please merge the objectives inside the introduction section instead of giving a separate heading for it.

Authors’ response: The study's objectives were integrated into the introduction section.

Comment 7: I apologize for not noticing it earlier but there is hardly any discussion on previous studies on this topic. Only first two paragraphs of introduction appear to mention a couple of them. It can be improved in two ways: 1) add a literature review section covering the previous studies on this topic, or 2) add these studies in the introduction section to develop the motivation, not only for this study but also for the project. It is quite possible that the project documents have mentioned these studies.

Authors’ response: The revised introduction now includes a comprehensive global literature review. As a result of adding new references and modifying existing ones, adjustments were made to the references in sections outside the Introduction to maintain consistency and accuracy throughout the document.

Comment 8: Please mention sources of the tables and figures at their bottom. 

Authors’ response: The tables and figures in this manuscript are derived directly from the raw data generated during the study. They are original and have not been sourced or quoted from any external references.

Comment 8: In the recommendations section, I suggest that authors should avoid stating the results. If these results are not mentioned in the results section, I suggest to move them to the results section otherwise simple suggest the key recommendations in this section. It is, in my opinion, too lengthy at the moment because it is stating results.

Authors’ response: It is revised to limit the inclusion of findings/ results in the recommendation section, as per the reviewer’s feedback.

---

## [Editor Report · Decision Letter 4]

13 Aug 2024

An evaluation of interventions within a Growth Through Nutrition project aimed at enhancing optimal nutrition and water, sanitation and hygiene (WASH) and nutrition practices among nutritionally most vulnerable households (MVHHs) in Ethiopia

PONE-D-23-39220R4

Dear Dr. Tirore,

We’re pleased to inform you that your manuscript has been judged scientifically suitable for publication and will be formally accepted for publication once it meets all outstanding technical requirements.

Kind regards,

Muhammad Ali, PhD

Academic Editor

PLOS ONE
---

## [Editor Report · Acceptance letter]

20 Aug 2024

PONE-D-23-39220R4 

PLOS ONE

Dear Dr. Tirore, 

I'm pleased to inform you that your manuscript has been deemed suitable for publication in PLOS ONE. Congratulations! Your manuscript is now being handed over to our production team.

Kind regards, 

on behalf of

Dr. Muhammad Ali 

Academic Editor

PLOS ONE